# Causality-Inspired Spatial-Temporal Explanations for Dynamic Graph Neural Networks

**Kesen Zhao**
City University of Hong Kong
Hong Kong, China
kesenzhao2-c@my.cityu.edu.hk

**Liang Zhang** *
Shenzhen Research Institute of Big Data
Guangdong, China
zhangliang@sribd.cn

## Abstract

Dynamic Graph Neural Networks (DyGNNs) have gained significant popularity in the research of dynamic graphs, but are limited by the low transparency, such that human-understandable insights can hardly be drawn from their predictions. Although a number of existing research have been devoted to investigating the interpretability of graph neural networks (GNNs), achieving the interpretability of DyGNNs is pivotally challenging due to the complex spatial-temporal correlations in dynamic graphs. To this end, we propose an innovative causality-inspired generative model based on structural causal model (SCM), which explores the underlying philosophies of DyGNN predictions by identifying the trivial, static, and dynamic causal relationships. To reach this goal, two critical tasks need to be accomplished including **(1)** disentangling the complex causal relationships, and **(2)** fitting the spatial-temporal explanations of DyGNNs in the SCM architecture. To tackle these challenges, the proposed method incorporates a contrastive learning module to disentangle trivial and causal relationships, and a dynamic correlating module to disentangle dynamic and static causal relationships, respectively. A dynamic VGAE-based framework is further developed, which generates causal-and-dynamic masks for spatial interpretability, and recognizes dynamic relationships along the time horizon through causal invention for temporal interpretability. Comprehensive experiments have been conducted on both synthetic and real-world datasets, where our approach yields substantial improvements, thereby demonstrating significant superiority.

## 1 Introduction

Dynamic graphs play a crucial role across a wide spectrum of real-world applications (Seo et al., 2018; You et al., 2018), including financial networks (Nascimento et al., 2021; Zhang et al., 2021), social networks (Berger-Wolf & Saia, 2006; Greene et al., 2010), and traffic networks (Peng et al., 2021; 2020). Unlike the widely studied static graphs, dynamic graphs can represent the spatial-temporal characteristics of real-world data, which is gaining great popularity in practical scenarios despite the high complexity (Pareja et al., 2020). Addressing the challenges posed by this complexity has led to the development of Dynamic Graph Neural Networks (DyGNNs) (Wang et al., 2023; Manessi et al., 2020; Beck et al., 2017; Zaki et al., 2016), which achieves significant advances in predictive tasks through accommodating the intricate interplay of spatial-temporal patterns.

Despite the aforementioned advantages, DyGNNS are usually limited by low transparency, such that human-understandable insights can hardly be drawn from their predictions. Existing works on the explanation of GNNs, such as GNNExplainer (Ying et al., 2019), XGNN (Yuan et al., 2020), and OrphicX (Lin et al., 2022) primarily focus on static networks. Therefore, these methods can hardly be directly adopted for the interpretability of dynamic networks due to the following two challenges induced by the complex spatial-temporal correlations in dynamic graphs. **1) Spatial interpretability**. The investigation of spatial interpretability critically relies on the extraction of subgraphs that can represent the characteristics of the complete graph in spatial dimension and elucidating outcomes in subsequent tasks. In essence, these subgraphs serve as substitutes for the

---

*Corresponding author.

original graphs, enabling the attainment of analogous results in downstream tasks. However, the sub-graph partition depends on the historical evolution over time as well as the spatial topology of the graph, which cannot be appropriately handled by the conventional methods designed for static graphs. **2) Temporal interpretability**. Temporal interpretability relies on the importance of representative sub-graphs over the time slots. In essence, it's essential to elucidate the significance of each time step concerning its impact on the outcomes of subsequent tasks. However, the temporal dynamics of a node are also impacted by its topological neighbor apart from its historical states. This makes it infeasible to directly adopt the techniques for time series research.

While causal inference provides an effective framework for investigating the interpretability in graph structures, a critical task to tackle these challenges is to disentangle the complicated types of relationships and accommodate them individually. To reach this goal, we propose an innovative causality-inspired spatial-temporal generative model via constructing the structural causal model (SCM) (Pearl, 2009) for dynamic graphs. In this line of research, previous works for static graphs categorize the relationships within the graph into trivial (or spurious) relationship and causal relationship (Lin et al., 2021; 2022), where trivial relationship captures the dispensable graph information while the causal relationship exploits the key information for tasks. Inspired by these works, we further divide the causal relationships in the SCM into static relationship and dynamic relationship to capture the complex spatial-temporal correlations. However, there remain two critical challenges to implementing such a SCM. The first challenge lies in the approach to disentangling the complex causal relationships as no explicit information is available for the identification of trivial, dynamic, and static relationships. The second challenge is the way to construct the SCM to fit the task of discovering spatial-temporal interpretability due to the lack of existing model for dynamic graphs.

To address these problems, we present a novel **Dy**namic **GNN Explainer**. Specifically, to disentangle the trivial relationship and the causal relationship, we propose a contrastive learning module to ensure the semantic similarity between the causal relationship and the original graph while enlarging the semantic distance between the causal relationship and the trivial relationship. To disentangle the dynamic relationship and the static relationship, we leverage the pre-trained target DyGNN model to guarantee the essential temporal correlation between neighboring subgraphs for the dynamic relationships and identify the rest independent causal information to the static relationships. We instantiate our DyGNNexplainer with a dynamic variational graph auto-encoder (VGAE) framework, which extracts the causal and dynamic relationships and maps them into a causal adjacency mask and a dynamic adjacency mask to accomplish the spatial explanation of graph-truth label. We further disentangle dynamic relationships along the time horizon by treating each subgraph as a causal invention and leverage the pre-trained target DyGNN model to measure its causal effect as the temporal explanation. To showcase the effectiveness of our approach, we generate synthetic dynamic datasets tailored for dynamic graph interpretability tasks, which fill the blank of dataset benchmarks and would facilitate further research in this domain. Experiments on both synthetic datasets and real-world datasets demonstrate superior performance of our method in both explanation tasks and real predictions. The code and the dataset benchmarks are available [1].

## 2 METHOD

### 2.1 PROBLEM STATEMENT

We denote a pre-trained target DyGNN model to be explained as $f = f_d \circ f_a$, where $f_a : \mathcal{G}_{1:T} \to \mathcal{R}$ is the aggregation function of DyGNN to capture temporal structures and feature patterns, $\mathcal{G}_{1:T}$ is the dynamic graph sets, $T$ is the total number of time steps, $\mathcal{R}$ is the aggregated high dimensional graph representation. $f_d : \mathcal{R} \to \mathcal{Y}$ is the downstream task function, which transforms graph representation to label space. $\mathcal{Y}$ is the final label prediction. Specifically, the input dynamic graph at the $t^{th}$ time step $\mathbf{G}_t = (\mathbf{X}_t, \mathbf{A}_t), t \in [1, T]$ includes the node attribute matrix $\mathbf{X}_t \in \mathbb{R}^{|V| \times D}$ and corresponding adjacency matrix $\mathbf{A}_t \in \mathbb{R}^{|V| \times |V|}$, where V is the set of nodes, D is the dimension of node attribute.

Explanation methods for DyGNNs aim to meet two critical criteria: *fidelity* and *interpretability*. *Fidelity* requires that a faithful explanation, represented by dynamic subgraphs, should align with how the target DyGNN behaves in the vicinity of the given dynamic graphs of interest (Ribeiro et al., 2016). In other words, when we feed the explanatory dynamic subgraphs to the target DyGNN, the

---

[1]https://github.com/kesenzhao/DyGNNExplainer

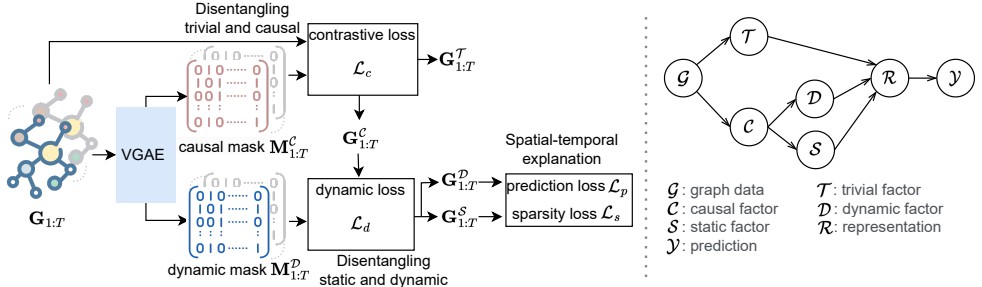

Figure 1: Overview of DyGNNExplainer and SCM for dynamic explanation.

outcomes should closely resemble those obtained from the original dynamic graphs. *Interpretability* (Pope et al., 2019) demands that generated explanations should highlight the most important parts of the input while disregarding the irrelevant components. Explanations for DyGNNs require both *spatial interpretability* and *temporal interpretability*. The spatial interpretability highlights the most important parts of graphs while temporal interpretability identifies the most important time slices. Moreover, an explainer should be versatile enough to explain any black-box model, adhering to the principle of being 'model-agnostic'.

Hence, our ultimate objective is to define a generative model, denoted as $\mathcal{F}$, to act as an explainer. This model should have the capability to pick out which aspects of the input contribute to the DyGNN prediction while meeting the fidelity and interpretability criteria outlined above. In line with prior research (Lin et al., 2021; Yuan et al., 2020; Lin et al., 2022), our focus lies in providing spatial-temporal explanations (dynamic subgraph set) for dynamic graph structures. We operate under the black-box setting, wherein we lack information regarding the ground-truth labels of the input graphs and do not require the inner workings of the target DyGNN's output generation process.

## 2.2 FRAMEWORK OVERVIEW

In this study, we propose a novel causality-inspired spatial-temporal DyGNNExplainer, shown in Figure 1. We first construct a Structural Causal Model (SCM) for dynamic graphs with trivial, dynamic, and static relationships, enabling a comprehensive understanding of dynamic graphs. Then we generate causal and dynamic soft masks, which enable backdoor adjustment, to intervene in the targeting causal and dynamic factors. After that, we employ a contrastive loss to separate trivial and causal relationships. To disentangle static and dynamic relationships, we employ a dynamic loss, which captures the temporal evolution within graphs and maintains independence from static information. Finally, we improve the generated static and dynamic explanations using prediction loss and sparsity loss, which enhances both prediction accuracy and spatial interpretability.

## 2.3 A CAUSAL VIEW ON DYGNNS

We first take a casual look at the DyGNN modeling and construct a Structural Causal Model (SCM) in Figure 1. It presents the causalities among seven variables: dynamic graph data $\mathcal{G}$, trivial relationship $\mathcal{T}$, causal relationship $\mathcal{C}$, dynamic relationship $\mathcal{D}$, static relationship $\mathcal{S}$, node representation $\mathcal{R}$, and prediction $\mathcal{Y}$, where links between variables represent cause-effect relationships. Here are some key explanations regarding SCM:

- $\mathcal{T} \leftarrow \mathcal{G} \rightarrow \mathcal{C}$: $\mathcal{C}$ represents the genuine causal relationship in dynamic graph data, while $\mathcal{T}$ signifies trivial relationships, often stemming from data biases or spurious patterns.

- $\mathcal{T} \rightarrow \mathcal{R} \leftarrow \mathcal{C}$: $\mathcal{R}$ is a high-dimensional representation of dynamic graph node data $\mathcal{G}$. To generate $\mathcal{R}$, the conventional strategy leverages both the trivial relationship $\mathcal{T}$ and the causal relationship $\mathcal{C}$ as inputs to extract discriminative information.

- $\mathcal{D} \rightarrow \mathcal{R} \leftarrow \mathcal{S}$: In dynamic graph $\mathcal{G}$, causal relationships $\mathcal{C}$ consist of dynamic relationship $\mathcal{D}$ and static relationship $\mathcal{S}$.

- $\mathcal{R} \rightarrow \mathcal{Y}$: The ultimate aim of dynamic graph representation learning is to predict graph properties, such as node or graph label.

From this SCM, we identify a backdoor path between $\mathcal{C}$ and $\mathcal{Y}$, i.e., $\mathcal{C} \leftarrow \mathcal{G} \rightarrow \mathcal{T} \rightarrow \mathcal{R} \rightarrow \mathcal{Y}$. In this path, the trivial relationship $\mathcal{T}$ acts as a confounder between $\mathcal{G}$ and $\mathcal{Y}$. Even if there's no direct link between $\mathcal{C}$ and $\mathcal{Y}$, the backdoor path can cause a misleading correlation between $\mathcal{C}$ and $\mathcal{Y}$, leading to incorrect predictions. Thus, it's crucial to block backdoor path to enable dynamic GNNs to effectively utilize causal relationships. Similarly, we have another backdoor path between $\mathcal{D}$ and $\mathcal{Y}$, i.e., $\mathcal{D} \leftarrow \mathcal{C} \rightarrow \mathcal{S} \rightarrow \mathcal{R} \rightarrow \mathcal{Y}$, and the static relationship $\mathcal{S}$ acts as confounder between $\mathcal{C}$ and $\mathcal{Y}$.

## 2.4 BACKDOOR ADJUSTMENT

Our research has emphasized the significance of safeguarding DyGNNs against confounding factors and distinguishing between dynamic and static relationships. This distinction is crucial for effectively utilizing causal relationships within dynamic graphs. Instead of modeling the confounded relationship denoted as $P(\mathcal{Y}|\mathcal{C})$ in Figure 1, our focus shifts towards graph representation learning that eliminates the backdoor path. Fortunately, we can achieve this by applying do-calculus principles to the variable $\mathcal{T}$. By doing so, we can estimate the probability distribution $P(\mathcal{Y}|do(\mathcal{C}))$ without interference from the confounder $\mathcal{T}$, utilizing standard backdoor adjustment. Similarly, we apply do-calculus to the variable $\mathcal{D}$ and estimate $P(\mathcal{Y}|do(\mathcal{D}))$ to eliminate the backdoor path caused by the confounder $\mathcal{S}$. Note that we can't directly employ the standard backdoor adjustment method due to confounder $\mathcal{T}$. To overcome this challenge, we merge the estimation of $P(\mathcal{Y}|do(\mathcal{C}))$ with that of $P(\mathcal{Y}|do(\mathcal{D}))$, resulting in the following equations:

$$P(\mathcal{Y}|do(\mathcal{D})) = \sum P(\mathcal{Y}|do(\mathcal{C}))P(\mathcal{S}) = \sum P(\mathcal{S}) \sum P(\mathcal{Y}|\mathcal{G})P(\mathcal{T}). \tag{1}$$

The first and second equations are based on the backdoor adjustment for confounder $\mathcal{S}$ and $\mathcal{T}$ respectively. Detailed derivation process is shown in Appendix A.4. However, there exist challenges for implementing backdoor adjustment. No explicit information is available for the identification of trivial, dynamic, and static relationships. To tackle the above challenges, we propose an effective method shown in the next subsection.

## 2.5 DISENTANGLING COMPLEX CAUSAL RELATIONSHIPS

Given a dynamic graph set $\mathbf{G}_{1:T} = (\mathbf{X}_{1:T}, \mathbf{A}_{1:T})$, we formulate the causal soft masks for causal relationship at $t^{th}$ time step as $\mathbf{M}_t^{\mathcal{C}} \in \mathbb{R}^{|V| \times |V|}$. Each element of the mask represents an attention score, typically falling within the range of $[0, 1]$. For arbitrary mask $\mathbf{M}$, we define $\overline{\mathbf{M}} = \mathbf{1} - \mathbf{M}$ as its complementary mask, where $\mathbf{1}$ is the all-one matrix. Consequently, we partition the entire dynamic graph set $\mathbf{G}_{1:T}$ into two distinct sets: causal set $\mathbf{G}_{1:T}^{\mathcal{C}} = (\mathbf{X}_{1:T}, \mathbf{A}_{1:T} \oplus \mathbf{M}_{1:T}^{\mathcal{C}})$ and trivial set $\mathbf{G}_{1:T}^{\mathcal{T}} = (\mathbf{X}_{1:T}, \mathbf{A}_{1:T} \oplus \overline{\mathbf{M}}_{1:T}^{\mathcal{C}})$. Where $\oplus$ is the element-wise dot product at each time step. Similarly, we formulate the dynamic soft masks as $\mathbf{M}_t^{\mathcal{D}} \in \mathbb{R}^{|V| \times |V|}$ to extract the dynamic relationships and its complementary mask $\overline{\mathbf{M}}_{1:T}^{\mathcal{D}}$ to extract the static relationships. Then, we have dynamic causal set $\mathbf{G}_{1:T}^{\mathcal{D}} = (\mathbf{X}_{1:T}, \mathbf{A}_{1:T} \oplus \mathbf{M}_{1:T}^{\mathcal{C}} \oplus \mathbf{M}_{1:T}^{\mathcal{D}})$ and static causal set $\mathbf{G}_{1:T}^{\mathcal{S}} = (\mathbf{X}_{1:T}, \mathbf{A}_{1:T} \oplus \mathbf{M}_{1:T}^{\mathcal{C}} \oplus \overline{\mathbf{M}}_{1:T}^{\mathcal{D}})$. Because the ground-truth trivial set, dynamic causal set, and static causal set are unavailable in real-world applications. We aim to capture the trivial, dynamic, and static relationships from the full graph by learning the masks.

**Estimating soft mask:** Inspired by the VGAE framework (Kipf & Welling, 2016), we proposed a dynamic VGAE-based encoder-decoder to estimate the soft masks of explainable subgraphs. We first consider the estimation of the causal soft mask matrix $\mathbf{M}_{1:T}^{\mathcal{C}}$. At the $t$-th time step, the causal soft mask matrix can be calculated as

$$\mathbf{M}_t^{\mathcal{C}} = f_v\left(\mathbf{X}_{1:t}, \mathbf{A}_{1:t}; \Theta_{\mathcal{C}}\right) = p(\mathbf{M}_t^{\mathcal{C}} \mid \mathbf{H}_t)q(\mathbf{H_t} \mid \mathbf{G}_{1:t}), \tag{2}$$

where $f_v$ is the encoder-decoder architecture with parameters $\Theta_{\mathcal{C}}$, $q(\cdot)$ is the encoder module, $p(\cdot)$ is the decoder module. The encoder utilizes posterior probabilities to encode the node embeddings into low-dimensional latent vector representations, which can be formulated as

$$q(\mathbf{H}_t \mid \mathbf{G}_{1:t}) = \Pi_{i=1}^N q\left(\mathbf{h}_{t,i} \mid \mathbf{G}_{1:t}\right), q\left(\mathbf{h}_{t,i} \mid \mathbf{G}_{1:t}\right) = \mathcal{N}\left(\mathbf{h}_{t,i} \mid \boldsymbol{\mu}_{t,i}, \mathrm{diag}\left(\boldsymbol{\sigma}_{t,i}^2\right)\right), \tag{3}$$

where $\mathbf{H}_t$ is the latent matrix. $\boldsymbol{\mu}_t$ and $\boldsymbol{\sigma}_t$ are means and variances of node latent embeddings learned by $GCN_\mu(\mathbf{G}_t)$ and $GCN_\sigma(\mathbf{G}_t)$ with different parameters. $\mathbf{h}_{t,i}$, $\boldsymbol{\mu}_{t,i}$ and $\boldsymbol{\sigma}_{t,i}$ are the $i^{th}$ column of $H_t$, $\boldsymbol{\mu}_t$ and $\boldsymbol{\sigma}_t$, respectively. We use the re-parameterization technique to avoid the problem that the

sampling operation cannot be back-propagated and updated by gradient descent. Then the decoder utilizes latent representations to generate the dynamic explainable subgraph as follows

$$p(\mathbf{M}_t^{\mathcal{C}} \mid \boldsymbol{H}_t) = \prod_{i=1}^{N} \prod_{j=1}^{N} p\left(M_{t,ij}^{\mathcal{C}} \mid \mathbf{h}_{t,i}, \mathbf{h}_{t,j}\right), p\left(M_{t,ij}^{\mathcal{C}} = 1 \mid \mathbf{h}_{t,i}, \mathbf{h}_{t,j}\right) = g\left(\mathbf{h}_{t,i}, \mathbf{h}_{t,j}\right) \quad (4)$$

where $M_{t,ij}^{\mathcal{C}}$ is the $i^{th}$ row and $j^{th}$ column of $\mathbf{M}_t^{\mathcal{C}}$ representing the probability of existence for edge $(v_t^i, v_t^j)$ at time $t$ in causal graph set and $g(\cdot, \cdot)$ calculates this probability.

According to the above encoder-decoder framework, the causal soft mask $\mathbf{M}_{1:T}^{\mathcal{C}}$ can be estimated based on the input of dynamic graph set $\mathbf{G}_{1:T}$. The soft causal mask assists in the formulation of the causal graph set $\mathbf{G}_{1:T}^{\mathcal{C}}$. According to the causal graph in Fig. 1, the static relationship $\mathcal{S}$ can also be treated as the co-founder between $\mathcal{D}$ and $\mathcal{Y}$, just like the trivial relationship $\mathcal{T}$ respect to $\mathcal{C}$ and $\mathcal{Y}$. Thus, we take the same VGAE-based encoder-decoder framework to generate the dynamic soft mask matrix, but with different parameters $\Theta_{\mathcal{D}}$, have the following estimating process:

$$\mathbf{M}_t^{\mathcal{D}} = f_v\left(\mathbf{X}_{1:t}, \mathbf{A}_{1:t} \oplus \mathbf{M}_{1:t}^{\mathcal{C}}; \Theta_{\mathcal{D}}\right). \quad (5)$$

Now we have the adjacency matrix of causal set $\mathbf{A}_{1:T}^{\mathcal{C}} = \mathbf{A}_{1:T} \oplus \mathbf{M}_{1:T}^{\mathcal{C}}$, dynamic causal set $\mathbf{A}_{1:T}^{\mathcal{D}} = \mathbf{A}_{1:T} \oplus \mathbf{M}_{1:T}^{\mathcal{C}} \oplus \mathbf{M}_{1:T}^{\mathcal{D}}$ and static causal set $\mathbf{A}_{1:T}^{\mathcal{S}} = \mathbf{A}_{1:T} \oplus \mathbf{M}_{1:T}^{\mathcal{C}} \oplus \overline{\mathbf{M}}_{1:T}^{\mathcal{D}}$. We need to disentangle the trivial, dynamic, and static relationships. We first disentangle the trivial relationship and the causal relationship via a contrastive learning method. Then, we propose a novel dynamic correlating method to disentangle the static relationship and dynamic relationship.

**Disentangling trivial and causal:** According to the criteria in Section 2.1, the explanation should meet fidelity. Since the causal subgraph set is the target and the trivial graph set can be treated as noise, the outcome of the target model with the causal subgraph set should be more like the original graph set, while the trivial subgraph set should be treated as negative ones. To disentangle the information in the trivial subgraph set and the causal subgraph set, we need to extract the embedding from them via some dynamic methods. Fortunately, our target is to provide fidelity and interpretability for the pre-trained DyGNN. Thus, we utilize the pre-trained aggregation function to obtain such embedding, which is then utilized to assist in disentangling trivial and causal information. During each time step $t$, the aggregation function $f_a(\cdot)$ would generate the embeddings via extracting the essential information until time $t$ for original graph set $\mathbf{G}_{1:t}$, causal subgraph set $\mathbf{G}_{1:t}^{\mathcal{C}}$ and trivial subgraph set $\mathbf{G}_{1:t}^{\mathcal{T}}$. More formally, we have

$$\mathbf{R}_t = f_a(\mathbf{G}_{1:t}), \mathbf{R}_t^{\mathcal{C}} = f_a(\mathbf{G}_{1:t}^{\mathcal{C}}), \mathbf{R}_t^{\mathcal{T}} = f_a(\mathbf{G}_{1:t}^{\mathcal{T}}). \quad (6)$$

The above outputs can be the node (graph) embedding for node (graph) classification tasks. We utilize contrastive learning to ensure the semantic similarity between the causal embedding $\mathbf{e}_t^{\mathcal{C}}$ and the original embedding $\mathbf{e}_t$ while enlarging the semantic distance between the causal embedding $\mathbf{e}_t^{\mathcal{C}}$ and the trivial embedding $\mathbf{e}_t^{\mathcal{T}}$. Then, we have the following contrastive loss:

$$\mathcal{L}_c = \frac{1}{T} \sum_{t=1}^{T} \log \frac{\exp\left(s(\mathbf{e}_t, \mathbf{e}_t^{\mathcal{C}})/\tau\right)}{\exp\left(s(\mathbf{e}_t, \mathbf{e}_t^{\mathcal{C}})/\tau\right) + \alpha_1 \exp\left(s(\mathbf{e}_t^{\mathcal{T}}, \mathbf{e}_t^{\mathcal{C}})/\tau\right) + \alpha_2 \sum_{k \neq t} \exp\left(s(\mathbf{e}_t^{\mathcal{T}}, \mathbf{e}_k^{\mathcal{C}})/\tau\right)} \quad (7)$$

where $\tau$ is the temperature coefficient, $s(\cdot, \cdot)$ measures the similarity, we utilize the dot product here.

**Disentangling static and dynamic:** According to the structure causal model in Fig. 1, the causal relationship can be further divided into the static relationship and the dynamic relationship. To extract the dynamic relationship and static relationship from the dynamic causal set $\mathbf{G}_{1:T}^{\mathcal{D}}$ and static causal set $\mathbf{G}_{1:T}^{\mathcal{S}}$, we utilize GCN with learn-able parameters $\Psi_{\mathcal{D}}$ and $\Psi_{\mathcal{S}}$.

$$\boldsymbol{H}_t^{\mathcal{D}} = GCN(\boldsymbol{A}_t^{\mathcal{D}}, \boldsymbol{X}_t; \Psi_{\mathcal{D}}), \boldsymbol{H}_t^{\mathcal{S}} = GCN(\boldsymbol{A}_t^{\mathcal{S}}, \boldsymbol{X}_t; \Psi_{\mathcal{S}}). \quad (8)$$

Dynamic relationships evolve over time steps but static relationships are independent across each time step. Specifically, the dynamic relationships in time step $t$ can be inferred from the history dynamic causal set $\mathbf{G}_{1:(t-1)}^{\mathcal{D}}$ while static information in time step $t$ is independent with history static causal graph set $\mathbf{G}_{1:(t-1)}^{\mathcal{S}}$. Formally, we have

$$\boldsymbol{H}_{1:(t-1)}^{\mathcal{D}} \longrightarrow \boldsymbol{H}_t^{\mathcal{D}}, \quad \boldsymbol{H}_{1:(t-1)}^{\mathcal{S}} \perp \boldsymbol{H}_t^{\mathcal{S}}. \quad (9)$$

Note that using the history dynamic relationship $\boldsymbol{H}^{\mathcal{D}}_{1:(t-1)}$ to predict the dynamic relationship $\boldsymbol{H}^{\mathcal{D}}_t$ directly is trivial since the GCN with parameters $\Psi_{\mathcal{D}}$ can map the graph set into embedding space without any useful information. Fortunately, we can use the pre-trained aggregation function $f_a(\cdot)$ again, which extracts the dynamic relationship from the original graph set. According to the fidelity criteria, the generated dynamic graph set should also guarantee that the pre-trained aggregation function can extract the dynamic relationship from it. Thus, we have the following dynamic loss:

$$\mathcal{L}_d = \frac{1}{T-1} \sum_{t=2}^{T} d(f_a\left(\mathbf{G}^{\mathcal{D}}_{1:(t-1)}\right), \boldsymbol{H}^{\mathcal{D}}_t), \tag{10}$$

where $d(\cdot, \cdot)$ measures the distribution distance. The dynamic loss guarantees that the dynamic relationships extracted from the dynamic causal graph set are highly correlated. The rest causal information would be independent, which identifies the static causal graph set. To make sure that the dynamic and static relationship can be disentangled as separately as possible, we would provide the sparsity constraints for the dynamic causal graph set shown later.

**Spatial-temporal explanation:** According to the structural causal model, both the dynamic relationship and the static relationship are the key to the prediction and should be utilized to predict the ground-truth label. The learned causal soft mask and the dynamic soft mask can assist in spatial interpretability which highlights the most important causal parts and dynamic causal parts of dynamic graphs. However, this is not enough for temporal interpretability. Due to the highly temporal correlation for dynamic relationships, it would be difficult to disentangle the dynamic relationship. To address the challenge, we treat the dynamic relationship at time $t$ as an invention. With the help of aggregation function $f_a(\cdot)$, we measure the causal effect of this invention via the change of embedding. Formally, we define the causal effect at time $t$ as follows:

$$\Delta \boldsymbol{H}^{\mathcal{D}}_t = f_a\left(\mathbf{G}^{\mathcal{D}}_{1:t}\right) - f_a\left(\mathbf{G}^{\mathcal{D}}_{1:(t-1)}\right). \tag{11}$$

We combine the causal effect for the dynamic relationship and the static relationship at time $t$ as the key causal information for $\mathbf{G}_t$ and propose the learn-able weight pooling method to aggregate all the information across all time slots as follows:

$$\boldsymbol{H}_T = \sum_{t=1}^{T} t_p(\Delta \boldsymbol{H}^{\mathcal{D}}_t \oplus \boldsymbol{H}^{\mathcal{S}}_t)\Delta \boldsymbol{H}^{\mathcal{D}}_t \oplus \boldsymbol{H}^{\mathcal{S}}_t, \quad t_p(\mathbf{H}) = Softmax(\Psi_{\mathcal{P}}\mathbf{H}/\|\Psi_{\mathcal{P}}\|), \tag{12}$$

where $\Psi_{\mathcal{P}}$ is the parameters to learn temporal importance $t_p(\Delta \boldsymbol{H}^{\mathcal{D}}_t \oplus \boldsymbol{H}^{\mathcal{S}}_t)$. $t_p(\Delta \boldsymbol{H}^{\mathcal{D}}_t \oplus \boldsymbol{H}^{\mathcal{S}}_t)$ provides importance of subgraphs over different time slots and thus assists in temporal interpretability. Based on pre-trained classifier $f_d(\cdot)$, we use aggregated embedding to explain the ground-truth label via prediction loss:

$$\mathcal{L}_p = l(f_d(\boldsymbol{H}_T), \mathcal{Y})), \tag{13}$$

where $l(\cdot, \cdot)$ is the entropy loss. To ensure human interpretability, the explained causal subgraph set should be sparsity. We take the sparsity requirement for both the causal graph set and the dynamic causal graph set via the sparsity loss:

$$\mathcal{L}_s = \sum_{t=1}^{T} \frac{\left\|\mathbf{A}^{\mathcal{C}}_t\right\|_1 + \left\|\mathbf{A}^{\mathcal{D}}_t\right\|_1}{\|\mathbf{A}_t\|_1}. \tag{14}$$

In summary, we learn the optimal explainable causal subgraphs, dynamic subgraphs, and temporal importance by solving the following optimization problems:

$$\min_{\Theta, \Psi} \mathcal{L}(\Theta, \Psi) = \lambda_1 \mathcal{L}_c + \lambda_2 \mathcal{L}_s + \lambda_3 \mathcal{L}_p + \lambda_4 \mathcal{L}_d \tag{15}$$

where $\Theta = \{\Theta_{\mathcal{C}}, \Theta_{\mathcal{D}}\}$, $\Psi = \{\Psi_{\mathcal{C}}, \Psi_{\mathcal{D}}, \Psi_{\mathcal{P}}\}$ and $\lambda_1$, $\lambda_2$, $\lambda_3$, and $\lambda_4$ are hyper parameters.

## 3 EXPERIMENTS

### 3.1 EXPERIMENTAL SETTINGS

**Datasets:** We utilize 4 synthetic datasets and 2 real-world datasets for the node classification task and the graph classification task. Table 1 shows the statistics of all datasets. Since our method is

Table 1: Statistics of datasets for both node and graph classification.

| Dataset | Node classification | | | | Graph classification | |
|---|---|---|---|---|---|---|
| | DBA-Shapes | DTree-Cycles | DTree-Grid | Elliptic | DBA-2motifs | MemeTracker |
| #nodes | 700 | 871 | 1,231 | 203,769 | 25,000 | 3.3 mil. |
| #edges | 4,110 | 1,950 | 3410 | 234,355 | 51,392 | 27.6 mil. |
| #labels | 7 | 3 | 3 | 2 | 3 | 2 |

Table 2: Explanation accuracy of different models (%). Where best performances are bold.

| Task | Dataset | GNNExplainer | PGExplainer | Gem | OrphicX | DyGNNExplainer |
|---|---|---|---|---|---|---|
| Node cls. | DBA-Shapes | 92.1 | 92.9 | 93.6 | 94.3 | **97.8\*** |
| | DTree-Cycles | 92.8 | 93.7 | 94.4 | 96 | **98.2\*** |
| | DTree-Grid | 85.2 | 85.9 | 87.1 | 90.5 | **94.2\*** |
| | Elliptic | 92.4 | 94.1 | 94.6 | 96.1 | **98.7\*** |
| Graph cls. | DBA-2motifs | 86.5 | 88.0 | 90.7 | 91.4 | **96.3\*** |
| | MemeTracker | 88.2 | 89.2 | 91.0 | 91.9 | **97.4\*** |

"**\***" indicates the statistically significant improvements (i.e., two-sided t-test with $p < 0.05$) over the best baseline. 'cls.' is short for classification.

the first study on dynamic graph interpretability, there are no directly available datasets suitable for the task of dynamic graph interpretability. So we dynamically transformed some commonly used static graph interpretability datasets. For node classification, we process three benchmark synthetic datasets BA-Shapes, Tree-Cycles, and Tree-Grid (Ying et al., 2019), as dynamic graph datasets DBA-Shapes, DTree-Cycles, and DTree-Grid. Furthermore, we utilize a real-world dynamic graph dataset Elliptic [2]. For graph classification, we process a benchmark synthetic dataset BA-2motifs dataset (Luo et al., 2020) as dynamic graph dataset DBA-2motifs. And we utilize a real-word dataset MemeTracker (Leskovec et al., 2009). More details of datasets and the generation process are shown in Appendix A.1.

**Baselines:** Since we are the first to explain DyGNNs, there are no existing dynamic graph interpretability benchmarks for comparison. And graph representation and graph generalization models are the target models we want to explain, there is no comparison between us and them. Consequently, we compare our approach against various powerful static interpretability frameworks for GNNs. They are GNNExplainer (Ying et al., 2019), PGExplainer (Luo et al., 2020), Gem (Lin et al., 2021), and OrphicX (Lin et al., 2022). For all these static baselines, we treat all nodes and edges as occurring simultaneously. More details about baselines and hyperparameter settings are shown in Appendix A.2.

## 3.2 RESULTS

**Explanation fidelity:** Explanation fidelity pertains to the accuracy of the explanations provided by various methods. To gauge this, we compare the predicted labels of explanatory subgraphs with the predicted labels of the original graphs as generated by the target model. For the static baselines, we simplify the target model by excluding temporal evolution. From the results presented in Table 2, our DyGNNExplainer surpasses all other baselines by a significant margin across all datasets, both for node classification and graph classification tasks. This underscores the fidelity of the explanations produced by our method, which, in contrast to static baselines, adeptly captures the intricate spatial-temporal correlations in dynamic graphs via our causality-inspired spatial-temporal structure. Causal-based methods OrphicX and Gem also outperform other baselines, affirming the effectiveness of causal inference in explanation tasks. However, these methods only differentiate between trivial and causal factors, disregarding the dynamic factor. Consequently, they fall short in explaining complex spatial-temporal relationships. In contrast, our method addresses this limitation by imposing two constraints.

**Explanation interpretability analysis:** Interpretability implies that the explainer should emphasize the most crucial components of the input data while disregarding irrelevant elements. In other words,

---

[2] http://www.kaggle.com/ellipticco/elliptic-data-set

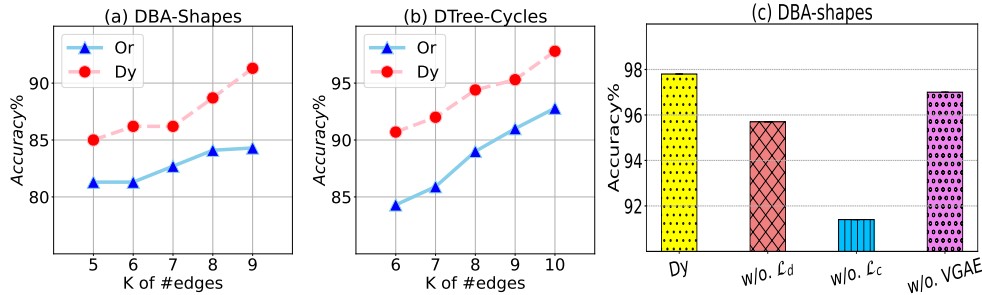

Figure 2: Interpretability analysis and ablation study. (a) Sparsity analysis on DBA-Shapes dataset (b) Sparsity analysis on DTree-Cycles dataset. (c) Ablation study on DBA-shapes. K is the edge number of each explanation subgraph. 'Or' is the OrphicX model, and 'Dy' is our DyGNNExplainer. 'w/o. $\mathcal{L}_d$', 'w/o. $\mathcal{L}_c$', and 'w/o. VGAE' are DyGNNExplainer without dynamic loss, contrastive loss, and VGAE, respectively.

Table 3: Prediction accuracy of different models (%). Where best performances are bold.

| Dataset | GNNExplainer | PGExplainer | Gem | OrphicX | Target | DyGNNExplainer |
|---|---|---|---|---|---|---|
| DBA-Shapes | 35.5 | 36.3 | 38.5 | 38.7 | 40.2 | **44.6\*** |
| Elliptic | 39.7 | 45.6 | 43.5 | 47.8 | 84.3 | **89.2\*** |

"**\***" indicates the statistically significant improvements (i.e., two-sided t-test with $p < 0.05$) over the best baseline.

the explanation subgraphs should exhibit a high degree of sparsity. To quantify this sparsity, we measure the number of subgraph edges (denoted as K). A smaller number of selected edges implies a higher sparsity. We compare the explanation accuracy of DyGNNExplainer with OrphicX on DBA-Shapes and DTree-Cycles, varying the number of edges (K) in the subgraph. As illustrated in Figure 2 (a) and (b), our method outperforms OrphicX with fewer edges in the subgraphs. This superiority arises from our model's adeptness at capturing spatial-temporal correlations in dynamic graphs, allowing it to encapsulate more critical information while ensuring interpretability.

**Prediction accuracy analysis:** While we aim to construct an explainer to faithfully elucidates the inner workings of the target model, it is equally imperative to ascertain the consistency of the interpreted results with real-world facts. Consequently, we also compare the prediction accuracy of our method with other baselines and target model on a synthetic dataset DBA-Shapes and a real-world dataset Elliptic for node classification task. As shown in the Table 3, DyGNNExplainer outperforms all other baselines by a substantial margin on both datasets. Particularly noteworthy is its 41.4% performance advantage over the best baseline, OrphicX, in the real-world Elliptic dataset. This underscores that our method generates explanations that align not only closely with the target model but also with real-world ground-truth. Static baselines, due to their inability to capture spatial-temporal correlations, fall short in accurately predicting the ground truth. Intriguingly, our model even surpasses the target model in terms of prediction accuracy. This is attributed to our model's ability to disentangle trivial, dynamic, and static relationships, thereby better capturing spatial-temporal correlations across graph time steps.

**Ablation study** We also delve into an ablation study on the DBA-Shapes dataset to dissect the roles of dynamic and static causal relationships. As depicted in Figure 2 (c), we present several variants for comparison. In 'w/o. VGAE', we replace the encoder with a simple GCN layer. Our findings reveal that DyGNNExplainer consistently outperforms the 'w/o. $\mathcal{L}_d$' and 'w/o. $\mathcal{L}_c$' versions. The 'w/o. $\mathcal{L}_d$' version, lacking the dynamic loss component, fails to effectively differentiate between dynamic and static factors, leading to an inability to capture temporal correlations across dynamic data. On the other hand, the 'w/o. $\mathcal{L}_c$' version experiences a significant drop in performance. This decline can be attributed to its inability to distinguish between causal and trivial factors, resulting in noisy independence that hinders the DyGNN explanations. These observations underscore the efficacy of our imposed constraints in disentangling trivial, dynamic, and static factors. Furthermore, DyGNNExplainer also outperforms the 'w/o. VGAE' version. This is primarily due to the superior capabilities of VGAE in harnessing spatial graph information to generate soft masks.

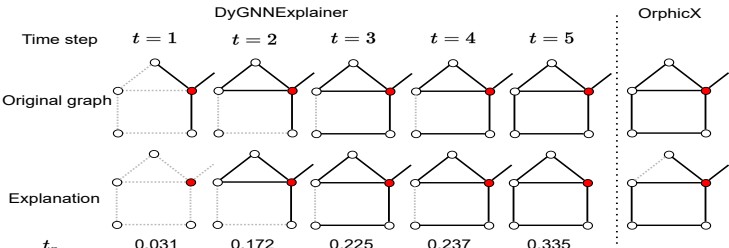

Figure 3: Case study on DBA-Shapes. The red node, designated as the target node for explanation, also serves as a connecting node between the 'house' motif and the base graph.

**Case study** To provide a more vivid demonstration of DyGNNExplainer's interpretability, we present a case study on DBA-Shapes dataset. As depicted in Figure 3, we visually represent both the original graph and the top six weighted edges of the generated causal subgraph across all time steps using DyGNNExplainer. Additionally, we compare our results with those obtained using OrphicX. To streamline the visualization, we have omitted edges and nodes that do not belong to the 'house' motif or are not directly linked to the target node. The variable $t_p$ denotes the importance of each time step. Notably, there is a gradual upward trend in $t_p$ values. This trend can be attributed to the increasing completeness of the 'house' motif in later time steps, rendering them more crucial for the final interpretation. The weight assigned to the first time slice is considerably lower than that of subsequent time steps. This discrepancy is due to the fact that the recognizable pattern of 'house' motif has not yet to fully materialize in the initial time step. Additionally, the weight for the fourth time step does not significantly surpass that of the previous one, as no new edges are added to the motif during this interval. This observation underscores the excellent temporal interpretability of our approach. Furthermore, DyGNNExplainer effectively identifies the 'house' motif from the original graph in the final time step, which explains the target node label. In contrast, OrphicX erroneously attributes an edge outside of the 'house' motif. This discrepancy vividly illustrates the superior spatial interpretability of our method.

## 4 RELATED WORK

A host of recent methods has emerged to provide explanations for Graph Neural Networks (GNNs) (Sui et al., 2022), focusing on identifying the most influential features (e.g., nodes, edges, or subgraphs) in input graphs to explain model predictions. These methods predominantly aim to generate input-dependent explanations. GNNExplainer (Ying et al., 2019) seeks soft masks for edges and node features through mask optimization to explain predictions. And Shokri et al. extend explanation methods designed for Convolutional Neural Networks (CNNs) to GNNs. However, these methods typically explain each instance individually and lack the ability to generalize graphs, limiting their global interpretability of the target model. Recognizing the issue of hindsight bias and the compromise of faithfulness when separately optimizing for each instance, PGExplainer (Luo et al., 2020) proposes learning a mask predictor for edge masks to provide explanations. XGNN (Yuan et al., 2020) focuses on investigating graph patterns leading to specific classes. In contrast to these approaches, our work leverages causality to achieve faithful explanations, distinguishing it from existing methods. More related work about causal inference is shown in Appendix A.3.

## 5 CONCLUSION

In conclusion, our work has addressed the critical challenges associated with interpretability in Dynamic Graph Neural Networks (DyGNNs). Our research has pioneered the development of DyGNN explanation, a novel approach tailored to the unique characteristics of dynamic graphs. Our experimental results, encompassing synthetic and real-world datasets, have demonstrated the superior performance of DyGNNExplainer in both explanation tasks and real predictions. Furthermore, we contribute to the field by generating synthetic dynamic datasets tailored for dynamic graph interpretability tasks, which lays the foundation for future developments in the field of dynamic graph analysis and interpretation.

ACKNOWLEDGEMENT

This work is supported by National Key R&D Program of China under Grant 2022YFA1003900, Internal Project of Shenzhen Research Institute of Big Data under Grant J00220230001

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

# A  APPENDIX

## A.1  DATASET

**Node classification datasets** We process three benchmark synthetic datasets BA-Shapes, Tree-Cycles, and Tree-Grid (Ying et al., 2019), as dynamic graph datasets DBA-Shapes, DTree-Cycles, and DTree-Grid. Furthermore, we utilize a real-world dynamic graph dataset Elliptic [3]. (1) BA-Shapes is a single graph consisting of a base Barabasi-Albert (BA) graph with 300 nodes and 80 'house'-structured motifs. These motifs are attached to randomly selected nodes from the BA graph. After that, random edges are added to perturb the graph. Node features are not assigned in BA-Shapes. (2) In the Tree-Cycles dataset, an 8-level balanced binary tree is adopted as the base graph. A set of 80 six-node cycle motifs are attached to randomly selected nodes from the base graph. (3) Tree-Grid is constructed in the same way as the Tree-Cycles, except that 3-by-3 grid motifs are used to replace the cycle motifs. (4) Elliptic is a Bitcoin transaction dataset. Nodes represent transactions and edges represent payment flows. We utilize this dataset for the node classification task.

**Graph classification datasets** We process a benchmark synthetic dataset BA-2motifs dataset (Luo et al., 2020) as dynamic graph dataset DBA-2motifs. And we utilize a real-word dataset Meme-Tracker (Leskovec et al., 2009). (1) BA-2motifs adopts the BA graphs as base graphs. Half graphs are attached with 'house' motifs and the rest are attached with five-node cycle motifs. Graphs are assigned to one of 2 classes according to the type of attached motifs. (2) MemeTracker contains 27.6 million news articles and blog posts from 3.3 million online sources with time stamps over a one-year period. The time-stamp indicates the information arrival time of a node from its parent nodes. We treat each website as a graph node, there is an edge between two nodes if a website forwards articles or blogs from another website. Thus, the propagation network forms a graph at a specific observation time stamp. All the graphs at different time stamps in a cascade have the same label.

**Dynamic data generation process** First, we randomly select a root vertex with a nonzero out-degree. The vertex is then added to the initially empty list and all outgoing edges are added to the initially empty first-in-first-out queue, where the first element added to the queue will be the first one to be removed. We choose an edge from the candidate set each time and calculate the time delay for the edge until the time delay exceeds a given time window (we use [0,1000] as the time window). When calculating the time delay of propagation, we follow a power-law distribution. Finally, we separate the graph with time delay of 100, 200, 300, 400, 500 (e.g., we form the first graph in a time-variant graph when the propagation time delay is 100). Thus, the average length in the synthetic data is 5 since we have 5 graphs in each time-variant graph. The whole data set is generated by repeating the above steps to generate 200 graph sequences. Then we relabel the nodes according to the diffusion direction of the motif nodes. For example, in the BA-Shapes dataset, if the node connected to the base in the house appears first, then other nodes in the motif appear, the nodes located at the top/middle/bottom of the 'house' are labeled with 1,2,3, respectively. In contrast, if other nodes in the motif appear first, the nodes located at the top/middle/bottom of the 'house' are labeled with 4,5,6. The nodes in the base graph are labeled with 0.

## A.2  EXPERIMENTAL SETTINGS

**Baselines** Since we are the first to explain DyGNNs, we compare our approach against various powerful static interpretability frameworks for GNNs.

- GNNExplainer (Ying et al., 2019) employs a technique called mask optimization to seek soft masks for edges and node features. These masks are instrumental in explaining the predictions made by GNNs.

- PGExplainer (Luo et al., 2020) takes a different approach by proposing the learning of a mask predictor. This predictor is responsible for generating edge masks, enabling instance-specific explanations for GNN predictions.

- Gem (Lin et al., 2021) distinguishes itself by focusing on explaining graph-structured data. It frames the explanation task for GNNs as a causal learning problem, aiming to generate concise subgraphs that contribute to predictions.

---

[3]http://www.kaggle.com/ellipticco/elliptic-data-set

- OrphicX (Lin et al., 2022) quantifies the causal attribution within the latent space without assuming the independence of explained features.

For all these static baselines, we input graphs without time attributes. That is, we treat all nodes and edges as occurring simultaneously.

**Hyperparameter setting** For the VGAE, we apply a two-layer GCN with output dimensions [32, 64, 128] and [16, 32, 64] in the encoder. The max time step T is set as 5. In the contrastive loss, the temperature coefficient $\tau$, weight parameters $\alpha_1$ and $\alpha_2$ are set from [0.2, 0.5, 0.8]. In the final optimization objects, the loss function weight parameters $\lambda_1$, $\lambda_2$, $\lambda_3$, and $\lambda_4$ are set from [0.2, 0.4, 0.6, 0.8, 1]. And the best performance is obtained where $\lambda_1 = 1$, $\lambda_2 = 0.4$, $\lambda_3 = 0.2$, and $\lambda_4 = 0.2$. We trained the explainers using the Adam optimizer (Kingma & Ba, 2014) with a learning rate of [1e-2, 1e-3, 1e-4] and batch size 64. When comparing to other baselines, we either utilize the same search range or adopt the optimal settings recommended by the original authors of the baselines. We divide the dataset as training set and test set with a ratio of 8:2, which is a common setting in previous works. All experiments are conducted on an NVIDIA Tesla V100S GPU, and the reported results are the average of three replicate experiments.

## A.3 MORE RELATED WORK ABOUT CAUSAL INFERENCE

The quest for DyGNN explanations often revolves around discerning the 'what if' and 'why' aspects, which inherently pertain to causality. Causal reasoning serves as a potent framework for addressing such inquiries (O'Shaughnessy et al., 2020). Various formalisms of causality have emerged, including structural causal models (SCM) (Pearl, 2009), Granger causality (Granger, 1969), and causal Bayesian networks (Pearl, 2009). While most explanation methods have been designed to elucidate conventional neural networks in image domains, Gem (Lin et al., 2021) represents a departure in its focus on explaining graph-structured data. Specifically, Gem formulates the task of explaining Graph Neural Networks (GNNs) as a causal learning problem. It proposes a causal explanation model capable of generating concise subgraphs that contribute to predictions. This approach fundamentally perturbs input aspects in the data space to observe the target GNN's response, naturally leading to the assumption of independent explained features. However, the interdependence inherent in graph-structured data, coupled with the non-linear transformations performed by GNNs, can compromise the efficacy and optimality of this assumption. Subsequently, OrphicX (Lin et al., 2022) quantifies the causal attribution of data aspects in the latent space without relying on the assumption of independent explained features. However, both Gem and OrphicX do not account for dynamic causal factors in dynamic graphs (Wu et al., 2022; Xia et al., 2023), thereby missing the opportunity to capture spatial-temporal correlations across dynamic graphs. In contrast, our approach differentiates between trivial, static, and dynamic causal factors, employing two constraints to ensure both spatial and temporal interpretability for DyGNNs.

## A.4 MORE DETAILED PROOFS AND EXPLANATIONS

**Backdoor adjustment** In Equation 1, we merge the estimation of $P(\mathcal{Y}|do(\mathcal{C}))$ with that of $P(\mathcal{Y}|do(\mathcal{D}))$. The detailed derivation process of backdoor adjustment can be shown as:

$$
\begin{aligned}
P(\mathcal{Y}|do(\mathcal{D})) &= \sum P(\mathcal{Y}|do(\mathcal{D}), \mathcal{S})P(\mathcal{S}|do(\mathcal{D})) \\
&= \sum P(\mathcal{Y}|do(\mathcal{C}))P(\mathcal{S}) \\
&= \sum P(\mathcal{S}) \sum P(\mathcal{Y}|do(\mathcal{C}), \mathcal{T})P(\mathcal{T}|do(\mathcal{C})) \\
&= \sum P(\mathcal{S}) \sum P(\mathcal{Y}|\mathcal{G})P(\mathcal{T}).
\end{aligned}
\tag{16}
$$

**Complexity analysis** VGAE-based encoder is the most time consuming component in our method. In the dynamic VGAE-based encoder, we generate the causal soft mask matrix $\mathbf{M}_t^{\mathcal{C}}$ and the dynamic soft mask matric $\mathbf{M}_t^{\mathcal{D}}$ with $O(d*|V|^2)$ complexity for T unique time steps, where d is the embedding size. The contrastive learning part also costs lots of time and has time complexity $O(|V| * d * T^2)$. Then, the total time complexity is $O(|V|^2 * d * T + |V| * d * T^2)$. Note that $T << |V|$, then we can ignore the time of contrastive learning and obtain the total time complexity as $O(|V|^2 * d * T)$.

