# OpenReview forum: "Causality-Inspired Spatial-Temporal Explanations for Dynamic Graph Neural Networks"
_ICLR.cc/2024/Conference — ICLR 2024 poster_

### Official Review · Reviewer_JCbW · 2023-10-26

**Soundness:** 2 fair
**Presentation:** 3 good
**Contribution:** 2 fair
**Rating:** 5
**Confidence:** 4

**Summary:**

This work aims at interpreting the Dynamic Graph Neural Networks (DyGNNs), and proposes an innovative causality-inspired
generative framework based on structural causal model (SCM), which explores the underlying philosophies of DyGNN predictions by identifying the trivial, static, and dynamic causal relationships. This work actually identifies the subgraph via a masking mechanism.

**Strengths:**

S1. The research motivation of this paper is clear. The existing studies on the interpretability of Dynamic Graph Neural Networks are still limited, thus it is meaningful to bridge such gap.

S2. The description of the existing challenges to the interpretability of DyGNNs is interesting. The authors argue that the first challenge lies in the approach to disentangling the complex causal relationships as no explicit information is available for the identification of trivial, dynamic, and static relationships.

**Weaknesses:**

W1. I'm confused about this sentence, 'Hence, our ultimate objective is to define a generative model', in Section 2.1. Throughout the whole paper, DyGNNExplainer is a representational model.

W2. Equation 1 should be described in detail. Since causal relationships ($C$) consist of dynamic ($D$) and static ($S$) relationships, there should exist $P(S) = P(C) - P(D)$. I guess this equation is derived from it, but I can't see the logical derivation.

W3. This paper still has not well addressed the interpretability issue of DyGNNs. The authors only provide evidence in performance improvements and static interpretability ('house' motif in BA-Shapes). We can not observe specific causal relationships in dynamic graphs from provided results. Besides, the baselines are interpretability methods conducted on static graphs, and the datasets are not classic dynamic graph datasets, such as traffic and citation network datasets. Thus, this proposed solution does not satisfy the expectation that exploring the interpretability of dynamic graph.

W4. Some important literature is missing, e.g., CauSTG for capturing invariant relations targets temporal shifts in ST graphs [1] and CaST for discovering via Structural Causal Model (SCM) with back-door and front-door adjustment [2]. The authors should distinguish the distinctions between the proposed DyGNNExplainer and  (CauSTG, CaST), especially the CaST.



[1] Xia Y, Liang Y, Wen H, et, al. Deciphering Spatio-Temporal Graph Forecasting: A Causal Lens and Treatment[J]. arXiv preprint arXiv:2309.13378, 2023.

[2] Zhou Z,  Huang Q,  Yang K, et, al. Maintaining the Status Quo: Capturing Invariant Relations for OOD Spatiotemporal Learning. In Proceedings of the 29th ACM SIGKDD Conference on Knowledge Discovery and Data Mining (KDD '23), 3603–3614.

**Questions:**

1. In Equation 7, $e$ is not defined in this paper, and the implementation of  $s( \cdot , \cdot )$ is also not provided.

2. Does $||A||$ operate by summing all elements of $A$ in Equation 14? Besides, does Equation 14 exist error? To satisying the sparsity requirement of causal and dynamic causal graph set, whether Equation 14 should be replaced by $\frac{{||A_t^C|{|_1} + ||A_t^S|{|_1}}}{{||{A_t}|{|_1}}}$?

3. In Table 2, OrphicX achieves the best performances on DTree-Grid (96.1). But, you bold your work DyGNNExplainer (94.2 < 96.1).

4. $\Theta $ should be replaced by $\Psi $ in the last line of Section 2.

5. How to interpret the 'dynamic' in DyGNNExplainer? Can the datasets in experiments support the augment raised in this paper, as it seems there are no dynamic graph in experiments?

6. Distinguish the distinctions between this work and CaST.

**Details Of Ethics Concerns:**

N/A.

---

> ### Author Response · Authors · 2023-11-22
> **Feedback to Reviewer # JCbW**
>
> We appreciate your overall positive assessment of our contributions and are grateful for your suggestion. We have already updated the new revision based on your suggestions.
>
> ------
>
> **Q1: Generative model**
>
> **W1:** Thanks for your question. Our dynamic graph interpretability task lies in providing spatial-temporal explanations (dynamic subgraph set) for dynamic graph structures. Consequently, our ultimate objective is to define a generative model, not a representational model. We make this more clear in Section 2.1.
>
> ------
>
> **Q2: Logical derivation of Equation 1**
>
> **W2:** In Equation 1, we merge the estimation of $P(\mathcal{Y}|do(\mathcal{C}))$ with that of $P(\mathcal{Y}|do(\mathcal{D}))$. The detailed derivation process of backdoor adjustment can be shown as:
>
> \begin{equation}
> \begin{aligned}
> P(\mathcal{Y}|do(\mathcal{D})) &= \sum P(\mathcal{Y}|do(\mathcal{D}), \mathcal{S})P(\mathcal{S}|do(\mathcal{D})) \\
> &= \sum P(\mathcal{Y}|do(\mathcal{C}))P(\mathcal{S})\\
> &= \sum P(\mathcal{S})\sum P(\mathcal{Y}|do(\mathcal{C}),\mathcal{T})P(\mathcal{T}|do(\mathcal{C})\\
> &= \sum P(\mathcal{S})\sum P(\mathcal{Y}|\mathcal{G})P(\mathcal{T}).
> \end{aligned}.
> \end{equation}
>
> We add this in Appendix A.4 in the revision.
>
> ------
>
> **Q3: Interpretability issue of DyGNNs**
>
> **W3:** Sorry for our unclear expression, all our experiments are conducted on dynamic datasets. In essence, our case study is conducted on DBA-Shapes to provide a more vivid demonstration of DyGNNExplainer's interpretability. We visually represent both the original graph and the top six weighted edges of the generated causal subgraph across all time steps using DyGNNExplainer.
>
> Regarding temporal interpretability, the scrutiny of variable $t_p$ reveals that DyGNNExplainer adeptly pinpoints the most critical time steps that significantly influence the target node label within the original graph.
>
> In terms of spatial interpretability, DyGNNExplainer proficiently discerns the presence of the 'house' motif within the original graph's concluding time step, offering a compelling explanation for the target node label.
>
> ------
>
> **Q4: Static datasets**
>
> **W4:** Since our method is the first study on dynamic graph interpretability, there are no directly available datasets suitable for the task of dynamic graph interpretability. So we dynamically transformed some commonly used static graph interpretability datasets. We add this in Section 3.1 in the revision.
>
> ------
>
> **Q5: Missing important literature**
>
> **W5:** You recommend some benchmarks for graph representation and graph generalization. But it is hard to compare our method with them since the task in our model is totally different from theirs. Graph representation and graph generalization models are the target models we want to explain. For interpretable methods, the target model is model-agnostic. The interpretability task is to provide good explanations for these models. We make this more clear in Section 3.1.
>
> Additionally, our methods of decoupling dynamic and static relationships are also different from CaST. To disentangle the static relationship and the dynamic relationship, we propose the constraint that dynamic relationships evolve over time steps but static relationships are independent across each time step. Consequently, we leverage the pre-trained target DyGNN model to guarantee the essential temporal correlation between neighboring subgraphs for the dynamic relationships and identify the rest independent causal information to the static relationships.
>
> ------
>
> **Q6: Implementation of Equation 7**
>
> **W6:** We utilize contrastive learning to ensure the semantic similarity between the causal embedding $\mathbf{e}^{\mathcal{C}}_t$ and the original embedding $\mathbf{e}_t$ while enlarging the semantic distance between the causal embedding $\mathbf{e}^{\mathcal{C}}_t$ and the trivial embedding $\mathbf{e}^{\mathcal{T}}_t$. $s(\cdot,\cdot)$ measures the similarity, we utilize the dot product here. We add the implementation in Section 2.5 in the revision.
>
> ------
>
> **Q7: Implementation of Equation 14**
>
> **W7:** $\left\|\mathbf{A}^{\mathcal{C}}_t\right\|_1$  and $\left\|\mathbf{A}^{\mathcal{D}}_t\right\|_1$ are $l_1$ norm of the causal graph set and the dynamic causal graph set, respectively. We sum up all elements here. The $\left\|\mathbf{A}^{\mathcal{S}}_t\right\|_1$  in Equation 14 should be replaced by $\left\|\mathbf{A}^{\mathcal{D}}_t\right\|_1$ , we have corrected it in revision.
>
> ------
>
> **Q8: Typo in Table 2 and Equation 15**
>
> **W8:** We are sorry that one of the data in Table 2 was filled in incorrectly. The best performance in DTree-Grid is obtained by DyGNNExplainer. We have corrected the typo in Table 2 in the revision. And we have corrected $\Theta$ with $\Psi$ in the last line of Section 2.
>
> ------
>
>  Thank you again for your constructive reviews. Hope that our response can address your concerns. We feel grateful for your appreciation.

---

> > ### Comment · Reviewer_JCbW · 2023-12-04
> > **Feedback to author rebuttal**
> >
> > The reviewer thanks the efforts made by authors on the detailed responses.  I still consider the name of DyGNN is doubtful and keep my score at this time.
> >
> >
> > Thanks.

---

### Official Review · Reviewer_fzQP · 2023-10-31

**Soundness:** 2 fair
**Presentation:** 1 poor
**Contribution:** 2 fair
**Rating:** 5
**Confidence:** 2

**Summary:**

This paper proposed a causality-inspired generative model to explain DyGNN predictions by identifying the trivial, static, and dynamic causal relationships. To experimentally evaluate the proposed approach, synthetic dynamic datasets are generated and provided. Evaluations on both synthetic datasets and real-world datasets demonstrate superior performance.

**Strengths:**

Originality: this paper is aimed at explaining dynamic graphs by proposing a causal inspired framework. Existing works on the explanation of GNNs are on static graphs. This paper instead focuses on dynamic graphs. Disentangling spatial and temporal relationships can be very challenging. This paper explicitly constructs a structural causal model by considering trivial relationships and causal relationships (consisting of static relationships and dynamic relationships) to solve this problem, which is interesting.

**Weaknesses:**

The presentation can be improved. It is hard for me to follow the paper well. For example, in the Introduction section, it is hard to straightforwardly understand the spatial interpretability and temporal interpretability. Illustrations can help readers understand better. Besides, it is not easy for me to understand the challenges for implementing the SCM (third paragraph in the Intro). Correspondingly, I didn’t see how the proposed approach addresses the challenges in the fourth paragraph.

The significance of the proposed approach is not clear. It is hard to judge the performance improvement achieved by DyGNNExplainer since other baselines are all for static graphs.

**Questions:**

In Table 2, for Node classification task, OrphicX performs better than DyGNNExplainer on DTree-Grid dataset but is not bolded?

Can you compare your model on static graphs to state-of-the-art explainers?

How sensitive the model is to the hyper parameters in Equation 15? What’s the computational complexity of solving Equation 15?

**Details Of Ethics Concerns:**

The link to the code and the dataset benchmark of this submission is **not anonymized**.

Post-rebuttal: this concern is addressed.

---

> ### Author Response · Authors · 2023-11-22
> **Feedback to Reviewer # fzQP**
>
> We appreciate your overall positive assessment of our contributions and are grateful for your suggestion. We have already updated the new revision based on your suggestions.
>
> ------
>
> **Q1: Problems in introduction section**
>
> **W2:** Thanks for your valuable questions and suggestions.
>
> **Spatial interpretability and temporal interpretability:**
>
> The investigation of spatial interpretability critically relies on the extraction of subgraphs that can represent the characteristics of the complete graph in spatial dimension and elucidate outcomes in subsequent tasks. In essence, these subgraphs serve as substitutes for the original graphs, enabling the attainment of analogous results in downstream tasks.
>
> Temporal interpretability relies on the importance of representative sub-graphs over the time slots. In essence, it's essential to elucidate the significance of each time step concerning its impact on the outcomes of subsequent tasks. We add this in Section 1.
>
> **Challenges for implementing the SCM:**
>
> The first challenge lies in the approach to disentangling the complex causal relationships as no explicit information is available for the identification of trivial, dynamic, and static relationships. The dynamic graph encapsulates intricate spatial-temporal relationships and dependencies, posing a challenge in directly conducting interventions on the confounder. This makes decoupling the trivial, dynamic, and static relationships very difficult.
>
> The second challenge is the way to construct the SCM to fit the task of discovering spatial-temporal interpretability due to the lack of existing models for dynamic graphs. As pioneers in the realm of dynamic graph interpretability, we encounter the challenge of lacking a causal model that can offer interpretive insights into intricate spatial-temporal relationships.
>
> **How the proposed approach addresses the challenges:**
>
> We propose two constraints to disentangle the trivial relationship, the static relationship, and the dynamic relationship, respectively.
>
> First, to disentangle the trivial relationship and the causal relationship, we propose that the causal relationship determines the downstream task results. Thus, we propose a contrastive learning module to ensure the semantic similarity between the causal relationship and the original graph while enlarging the semantic distance between the causal relationship and the trivial relationship.
>
> Second, to disentangle the static relationship and the dynamic relationship, we propose that dynamic relationships evolve over time steps but static relationships are independent across each time step. Consequently, we leverage the pre-trained target DyGNN model to guarantee the essential temporal correlation between neighboring subgraphs for the dynamic relationships and identify the rest independent causal information to the static relationships.
>
> Thank you again for your questions. It is also our goal to clean up our motivations and solutions. We aspire that through our explanations, you gain a more clear understanding of our work.
>
> ------
>
> **Q2: Static datasets**
>
> **W2:** Since our method is the first study on dynamic graph interpretability, there are no directly available datasets suitable for the task of dynamic graph interpretability. So we dynamically transformed some commonly used static graph interpretability datasets. We add this in Section 3.1 in the revision.
>
> ------
>
> **Q3: Baselines**
>
> **W3:** Since our method is the first study on dynamic graph interpretability, we can only utilize the baselines from static graph interpretability tasks. To achieve a relatively fair comparison, we add the time factor into baselines so as to make them suitable for dynamic graph interpretability. Dynamic graphs are much more complex than static graphs due to their spatial-temporal correlations. It is meaningless if we compare our method with baselines on static graphs.
>
> ------
>
> **Q4: Typo in Table 2**
>
> **W4:** We are sorry that one of the data in Table 2 was filled in incorrectly. The best performance in DTree-Grid is obtained by DyGNNExplainer. We have corrected the typo in Table 2 in the revision.
>
> ------

---

> > ### Author Response · Authors · 2023-11-22
> > **Feedback to Reviewer # fzQP**
> >
> > **Q5: Hyperparameters and computational complexity in Equation 15**
> >
> > **W5:** Thanks for your suggestions. We supplement the parameter search range and optimal parameter selection of the four constraints of the loss function to help improve the reliability and reproducibility of the results.
> >
> > In the final optimization objects, the loss function weight parameters $\lambda_1$, $\lambda_2$, $\lambda_3$, and $\lambda_4$ are set from [0.2, 0.4, 0.6, 0.8, 1]. And the best performance is obtained where $\lambda_1=1$, $\lambda_2=0.4$, $\lambda_3=0.2$, and $\lambda_4=0.2$. We add this in Appendix A.2.
> >
> > VGAE-based encoder is the most time consuming component in our method. In the dynamic VGAE-based encoder, we generate the causal soft mask matrix $\mathbf{M}^{\mathcal{C}}$ and the dynamic soft mask matric $\mathbf{M}^{\mathcal{D}}$ with $O(d*|V|^2)$ complexity for T unique time steps, where d is the embedding size. The contrastive learning part also costs lots of time and has time complexity O($|V|$*$d$*$T^2$). Then, the total time complexity is O($|V|^2$*$d$*$T$ + $|V|$*$d$*$T^2$). Note that $T << |V|$, then we can ignore the time of contrastive learning and obtain the total time complexity as O($|V|^2$*$d$*$T$).
> >
> > ------
> >
> > **Q6: Codebase link**
> >
> > **W6:** So sorry we attached the wrong code link. The wrong code link is unrelated to this paper and does not include any information about authors. We have updated the correct anonymous code link in the new version.
> >
> > ------
> >
> > Thank you again for your constructive reviews. Hope that our response can address your concerns. We will feel grateful if you could boost our paper.

---

### Official Review · Reviewer_U7py · 2023-11-03

**Soundness:** 3 good
**Presentation:** 3 good
**Contribution:** 3 good
**Rating:** 8
**Confidence:** 2

**Summary:**

This paper proposes a novel approach for interpretability in dynamic graph neural networks. The proposed framework is demonstrated on both synthetic and real-world datasets. The experimental results show that the proposed method outperforms the baselines (all baselines are for explaining static graph neural networks). Another contribution is that the paper constructs a new synthetic benchmark dataset for dynamic graph interpretability tasks.

**Strengths:**

The proposed framework is the first work for interpretability in dynamic graph neural networks. This is a significant contribution. The paper is well organized and clearly described. The method is technically sound. The experiments are comprehensive and the results show the effectiveness of the proposed method. The new constructed benchmark dataset is a good addition to the research domain.

**Weaknesses:**

Minors:
In Figure 1, the text is too small.

**Questions:**

In table 2, the best performance for OrphicX is obtained by DTree-Grid?

---

> ### Author Response · Authors · 2023-11-22
> **Feedback to Reviewer # U7py**
>
> We appreciate your overall positive assessment of our contributions and are grateful for your suggestion. We have already updated the new revision based on your suggestions.
>
> ------
>
> **Q1: Text in Figure 1**
>
> **W2:** We have enlarged the text in Figure 1in the new version.
>
> ------
>
> **Q2: Typo in Table 2**
>
> **W2:** We are sorry that one of the data in Table 2 was filled in incorrectly. The best performance in DTree-Grid is obtained by DyGNNExplainer. We have corrected the typo in Table 2 in the revision.
>
> ------
>
> Thank you again for your constructive reviews. Hope that our response can address your concerns. We feel grateful for your appreciation.

---

### Official Review · Reviewer_8sLV · 2023-11-06

**Soundness:** 3 good
**Presentation:** 3 good
**Contribution:** 3 good
**Rating:** 6
**Confidence:** 4

**Summary:**

This paper presents a causal approach to improving the interpretability of GNNs. The authors have integrated a contrastive learning module that distinguishes between non-causal and causal relationships, enhancing the clarity of the model's decision-making process. Additionally, a dynamic correlating component is employed to differentiate dynamic from static causal relationships, providing a nuanced understanding of changes over time. Furthermore, the authors utilize a VGAE-based model to generate causal-and-dynamic masks, which contribute to spatial interpretability. This model also captures dynamic relationships across temporal scales through causal inference, thereby boosting the model's ability to interpret temporal data.

**Strengths:**

(1) The composition and articulation of the paper are logical and coherent. The use of a causality-driven approach to enhance the out-of-distribution generalization capabilities of dynamic GNNs is intriguing.

(2) Introducing research on temporal distribution shift in sequential processes is important and may provide valuable insights for subsequent studies.

**Weaknesses:**

(1) The paper's presentation appears problematic, particularly in the description of the backdoor adjustment. While simplified results are provided in the main text, the specific derivation process is absent and should be relegated to the appendix. Additionally, the computational intensity of introducing temporal masks, which could be exacerbated by the incorporation of contrastive learning (VGAE is known to be computationally demanding), is not addressed. The authors should include complexity descriptions to inform the reader. However, these issues are not discussed in the paper.

(2) The proposal of 4 loss functions can be unfriendly to network training. If even one parameter is improperly tuned, it could lead to significant instability or even failure in network training. The authors should systematically discuss parameter selection techniques or guidelines to aid those who follow in this line of work.

(3) There is a lack of related experiments: although experiments are conducted, there is a shortage of benchmarks in this field. It is recommended that the authors refer to [1] to add more experiments to validate the effectiveness of their DyGNN, such as including the Ogbn-Arxiv dataset. Additionally, an ablation study replacing VGAE-like models is crucial to help others understand the contribution of each model component.

(4) Related work is missing from the paper, especially concerning spatio-temporal related work [5], generalization/extrapolation on graphs, and causality learning [2-4]. The authors should consider these areas to provide a more comprehensive context for their research.

[2] Sui, Yongduo, et al. "Causal attention for interpretable and generalizable graph classification." Proceedings of the 28th ACM SIGKDD Conference on Knowledge Discovery and Data Mining. 2022.

[3] Wu, Ying-Xin, et al. "Discovering invariant rationales for graph neural networks." arXiv preprint arXiv:2201.12872 (2022).

[4] Miao, Siqi, Mia Liu, and Pan Li. "Interpretable and generalizable graph learning via stochastic attention mechanism." International Conference on Machine Learning. PMLR, 2022.

[5] Xia, Yutong, et al. "Deciphering Spatio-Temporal Graph Forecasting: A Causal Lens and Treatment." arXiv preprint arXiv:2309.13378 (2023).

**Questions:**

See weakness

---

> ### Author Response · Authors · 2023-11-22
> **Feedback to Reviewer #8sLV**
>
> We appreciate your overall positive assessment of our contributions and are grateful for your suggestion. We have already updated the new revision based on your suggestions.
>
> ------
>
> **Q1: Derivation process and complexity descriptions**
>
> **A1:** In Equation 1, we merge the estimation of $P(\mathcal{Y}|do(\mathcal{C}))$ with that of $P(\mathcal{Y}|do(\mathcal{D}))$. The detailed derivation process of backdoor adjustment can be shown as:
> $$
> P(\mathcal{Y}|do(\mathcal{D})) = \sum P(\mathcal{Y}|do(\mathcal{D}), \mathcal{S})P(\mathcal{S}|do(\mathcal{D})) \\
> = \sum P(\mathcal{Y}|do(\mathcal{C}))P(\mathcal{S})\\
> = \sum P(\mathcal{S})\sum P(\mathcal{Y}|do(\mathcal{C}),\mathcal{T})P(\mathcal{T}|do(\mathcal{C})\\
> = \sum P(\mathcal{S})\sum P(\mathcal{Y}|\mathcal{G})P(\mathcal{T}).
> $$
>
> VGAE-based encoder is the most time consuming component in our method. In the dynamic VGAE-based encoder, we generate the causal soft mask matrix  $\mathbf{M}^{\mathcal{C}}$ and the dynamic soft mask matric $\mathbf{M}^{\mathcal{D}}$ with $O(d*|V|^2)$ complexity for T unique time steps, where d is the embedding size. The contrastive learning part also costs lots of time and has time complexity O(|V|*d*$T^2$). Then, the total time complexity is O($|V|^2$*d*T + |V|*d*$T^2$). Note that $T << |V|$, then we can ignore the time of contrastive learning and obtain the total time complexity as O(|V|^2*d*T).
>
> We add this in Appendix A.4  in the new version.
>
> ------
>
> **Q2: Loss function hyperparameters**
>
> **A2:** Thank you for your valuable suggestions. In graph interpretation tasks, they usually have many constraints and require different types of loss functions to satisfy these constraints, e.g., OrphicX has 4 loss functions for the static graph interpretation. We supplement the parameter search range and optimal parameter selection of the four constraints of the loss function to help improve the reliability and reproducibility of the results.
>
> In the final optimization objects, the loss function weight parameters $\lambda_1$, $\lambda_2$, $\lambda_3$, and $\lambda_4$ are set from [0.2, 0.4, 0.6, 0.8, 1]. And the best performance is obtained where $\lambda_1=1$, $\lambda_2=0.4$, $\lambda_3=0.2$, and $\lambda_4=0.2$. We add this in Appendix A.2.
>
> Once again, I extend my gratitude for your invaluable suggestions. It remains our primary objective to deliver trustworthy interpretive results.
>
> ------
>
> **Q3: Shortage of benchmarks and ablation study**
>
> **A3:** Thank you for your valuable questions. Although there have been some works on static graph interpretability, to the best of our knowledge, we are the first to study dynamic graph interpretability. Therefore, we have no existing dynamic graph interpretability benchmarks for comparison, and can only compare with currently competitive static interpretability methods.
>
> On the other hand, you recommend some benchmarks for graph representation and graph generalization. But it is hard to compare our method with them since the task in our model is totally different from theirs. Graph representation and graph generalization models are the target models we want to explain. For interpretable methods, the target model is model-agnostic. The interpretability task is to provide good explanations for these models. We make this more clear in Section 3.1.
>
> Given the novelty of our study on dynamic graph interpretability, there is a dearth of readily available synthetic datasets suitable for this purpose. Consequently, we've undertaken dynamic transformations of commonly used synthetic static graph interpretability datasets such as BA-Shapes. We also use two real-world dynamic graph datasets, Elliptic for node classification and MemeTracker for graph classification. The Ogbn-Arxiv dataset is a real-world data set but a static graph dataset. If we dynamically transform it, it will lose its real meaning. We make this more clear in Section 3.1.
>
> For ablation study, we provide a DyGNNExplainer version without VGAE ('w/o. VGAE'). In this version, we replace the encoder with a simple GCN layer. DyGNNExplainer outperforms the `w/o. VGAE' version. This is primarily due to the superior capabilities of VGAE in harnessing spatial graph information to generate soft masks. We provide this ablation study in Section 3.2.
>
> ------
>
> **Q4: Related work**
>
> **A4:** Thank you for your valuable suggestions. Although our interpretability task is totally different from the graph generalization task, their method is the model we want to explain and is helpful in our work. Additionally, the spatial-temporal method and the causal method are directly related to our tasks. We add these related works in Section 4 and Appendix 4.3.
>
> ------
>
> Thank you again for your constructive reviews. Hope that our response can address your concerns. We will feel grateful if you could boost our paper.

---

> > ### Comment · Reviewer_8sLV · 2023-11-23
> > **Official Comment by Reviewer 8sLV**
> >
> > Sorry for the late reply. Thank you for the detailed response from the author; the major concern has already been addressed. I will raise my score to acceptance, I believe that with the added content, this work has become clearer and more coherent, especially the comparative analysis of related work, which has resolved my questions effectively.

---

### Author Response · Authors · 2023-11-22
**General response**

We thank all reviewers for their insightful comments and suggestions. We are particularly encouraged by the reviewers’ feedback. We have made a heavy revision to our paper according to the reviewer's constructive suggestions. Below we summarize some key modifications in this revision:

-  Derivation process of backdoor adjustment in Equation 1in Appendix A.4 (Reviewer #1, Reviewer #4)
-  Complexity descriptions of dynamic VGAE in Appendix A.4 (Reviewer #1)
-  Loss function hyperparameters in Appendix A.2 (Reviewer #1, Reviewer #3)
-  Dataset and baseline selection in Section 3.1 (Reviewer #1, Reviewer #3, Reviewer #4)
-  More related works in Section 4 and Appendix 4.3 (Reviewer #1)
-  Clearer explanation and presentation
   -  Ablation study in Section 3.2 (Reviewer #1)
   -  Text in Figure 1 (Reviewer #2)
   -  Motivations and solutions in Section 1 (Reviewer #3, Reviewer #4)
   -  Graph interpretability task in Section 2.1 (Reviewer #4)
   -  Implementation of Equation 7 (Reviewer #4)


Moreover, we have carefully checked our paper typos and reorganized the presentation. If there still remains any consideration, please kindly let us know. We are very happy to make a further revision in light of your great suggestions. We will address comments by each of the reviewers individually.

---

### Meta-Review · Area_Chair_qM9T · 2023-12-06

**Metareview:**

This paper presents a causal approach to improving the interpretability of Graph Neural Networks (GNNs). It leverages a contrastive learning module that distinguishes between non-causal (static and trivial) and causal relationships. The model also captures dynamic relationships across temporal scales through causal inference, thus enhancing its ability to interpret temporal data.

While reviewers agree that the methodology is novel, they raise concerns regarding the validation pipeline, which seems to focus primarily on static graphs. Although the authors hand-crafted a synthetic dataset, it failed to alleviate these concerns for some reviewers. Additionally, concerns remain about whether the methods adequately address the core interpretability challenges of dynamic GNNs (DyGNNs).

In a nutshell, the method is novel and the results are compelling in both synthetic and real-world, albeit mostly static, datasets, which heavily nuance the results. I would recommend to accept.

**Justification For Why Not Higher Score:**

The reviewers comment that most of the benchmarks are in static datasets, so it is not clear how the method would actually behave in dynamical graphs.

**Justification For Why Not Lower Score:**

The framework it is interesting and novel. The results are compelling in static graphs.

---

### Decision · Program_Chairs · 2024-01-16

Accept (poster)